# Carboxymethyl-Dextran-Coated Superparamagnetic Iron Oxide Nanoparticles for Drug Delivery: Influence of the Coating Thickness on the Particle Properties

**DOI:** 10.3390/ijms232314743

**Published:** 2022-11-25

**Authors:** Chiara Turrina, Davide Milani, Anna Klassen, Diana M. Rojas-González, Jennifer Cookman, Matthias Opel, Barbara Sartori, Petra Mela, Sonja Berensmeier, Sebastian P. Schwaminger

**Affiliations:** 1Chair of Bioseparation Engineering, School of Engineering and Design, Technical University of Munich, 85748 Garching, Germany; 2Chair of Medical Materials and Implants, Department of Mechanical Engineering, Munich Institute of Biomedical Engineering, TUM School of Engineering and Design, Technical University of Munich, 85748 Garching, Germany; 3Department of Chemical Sciences, Bernal Institute, University of Limerick, Castletroy, V94 T9PX Limerick, Ireland; 4Walther-Meißner-Institut, Bayerische Akademie der Wissenschaften, 85748 Garching, Germany; 5Institute of Inorganic Chemistry, Graz University of Technology, Stremayrgasse 9/IV, 8010 Graz, Austria; 6Division of Medicinal Chemistry, Otto Loewi Research Center, Medical University of Graz, Neue Stiftingtalstraße 6, 8010 Graz, Austria; 7BioTechMed-Graz, 8010 Graz, Austria

**Keywords:** carboxymethyl dextran, iron oxide nanoparticles, antimicrobial peptide, magnetically controlled drug delivery, agglomeration behavior

## Abstract

Carboxymethyl-dextran (CMD)-coated iron oxide nanoparticles (IONs) are of great interest in nanomedicine, especially for applications in drug delivery. To develop a magnetically controlled drug delivery system, many factors must be considered, including the composition, surface properties, size and agglomeration, magnetization, cytocompatibility, and drug activity. This study reveals how the CMD coating thickness can influence these particle properties. ION@CMD are synthesized by co-precipitation. A higher quantity of CMD leads to a thicker coating and a reduced superparamagnetic core size with decreasing magnetization. Above 12.5–25.0 g L^−1^ of CMD, the particles are colloidally stable. All the particles show hydrodynamic diameters < 100 nm and a good cell viability in contact with smooth muscle cells, fulfilling two of the most critical characteristics of drug delivery systems. New insights into the significant impact of agglomeration on the magnetophoretic behavior are shown. Remarkable drug loadings (62%) with the antimicrobial peptide lasioglossin and an excellent efficiency (82.3%) were obtained by covalent coupling with the EDC/NHS (N-ethyl-N′-(3-(dimethylamino)propyl)carbodiimide/N-hydroxysuccinimide) method in comparison with the adsorption method (24% drug loading, 28% efficiency). The systems showed high antimicrobial activity with a minimal inhibitory concentration of 1.13 µM (adsorption) and 1.70 µM (covalent). This system successfully combines an antimicrobial peptide with a magnetically controllable drug carrier.

## 1. Introduction

Nanomedicine has introduced novel therapeutic and diagnostic capabilities that can address previously inaccessible issues in medicine, e.g., targeted delivery or improved medical imaging [1]. As a result of their simple and inexpensive production, good biocompatibility, high surface area to volume ratio, and superparamagnetic behavior, functionalized iron oxide nanoparticles (IONs) are ideal for applications in nanomedicine [2,3]. Their properties render them appropriate for use as T2 contrast agents in magnetic resonance imaging (MRI), cancer therapy such as hyperthermia treatment or inhibitory factor replacement, and even as carriers in magnetically controlled drug delivery systems [4,5,6,7,8]. The implementation of IONs as drug carriers for the targeted drug delivery of, e.g., anticancer (doxorubicin) or antimicrobial drugs (lasioglossin (LL)) increases the efficacy by improving the bioavailability of the drug and reduces the dose and associated systemic toxicity, including undesirable side effects [1,9,10,11]. IONs can be taken up by a cell through adsorptive endocytosis or by the macrophages. The binding of a drug, e.g., a peptide, to IONs can improve its cellular uptake [12]. When developing a new nanomedicine product such as a drug delivery system, a complete physicochemical characterization of the product must be carried out and be made readily available [13,14,15]. In addition, the biocompatibility and pharmacokinetic or pharmacodynamic behavior of the nanomedicine must be known before its use in the human body and in therapy [16,17,18]. The European Medicines Agency (EMA) has emphasized the importance of these criteria for developing new nanomedicines to ensure and simplify the development and approval process [1,19].

Magnetite (Fe_3_O_4_) is one of the variants of iron oxide found in nature and is often used for applications in nanomedicine due to its superparamagnetic behavior, high specific surface area, and biocompatibility [20]. The most common synthesis route used to produce iron oxide nanoparticles is co-precipitation using the Massart process (Equation (1)) [2,20,21]:(1)$${\mathrm{Fe}}^{2+}+2{\mathrm{Fe}}^{3+}+8{\mathrm{OH}}^{-}\;\to \;{\mathrm{Fe}(\mathrm{OH})}_{2}\;+\;2{\mathrm{Fe}(\mathrm{OH})}_{3}\;\to \;{\mathrm{Fe}}_{3}O_{4}\;+\;4H_{2}O$$

Magnetite is unstable in an oxygen atmosphere and can quickly oxidize to maghemite (γ-Fe_2_O_3_), leading to a decrease in the saturation magnetization [22,23]. Furthermore, uncoated iron oxide cores tend to agglomerate over time due to surface energy minimization, thus reducing the dispersibility and usable surface area [20]. Agglomerates of >100 nm may be formed, which impede the application in biomedical fields [23,24,25]. Bare IONs (BIONs) can form non-specific interactions with blood serum proteins [22,26]. This interaction can reduce the half-life of the particles in the body due to their opsonization and subsequent rapid removal from the bloodstream [27,28,29]. The abovementioned disadvantages limit the usability of BIONs in the field of drug delivery.

A solution to these problems is the method of coating with organic or inorganic polymers [22,23]. Carboxymethyl dextran (CMD) is an interesting coating for drug delivery applications for the following reasons. ION@CMD can be easily synthesized by an in situ co-precipitation (Figure 1) [21,30,31].

The Food and Drug Administration has already approved IONs with a modified dextran coating in combination with the drugs Ferumoxytol and Feridex IV for the treatment of anemia and as a contrast agent in MRI [32,33,34]. The variant of dextran is increasingly used as a coating due to its high density of free carboxyl groups [35,36]. The carboxyl groups are efficient tools for cross-linking the particles with primary amines in therapeutic proteins or peptides. Li et al. exploited this hypothesis by covalently binding anti-BSA antibodies to the particle surface via EDC/NHS (N-ethyl-N′-(3-(dimethylamino)propyl)carbodiimide/N-hydroxysuccinimide) activation [37]. Vasic et al. immobilized alcohol dehydrogenases (ADH) onto the particle surface without activity loss [30]. In previous studies of CMD-coated IONs, the negative surface charge led to a slower clearance from the bloodstream and less fouling activity and promoted the cellular uptake in Caco2-cells [35,36,38]. Furthermore, the negatively charged coated IONs enable the adsorption of positively charged drugs, such as antimicrobial peptides (AMP) [31].

AMPs, with their cationic properties, can accumulate on the negatively charged membrane surface of bacteria and disrupt the structure of the lipid bilayer. Therefore, cytoplasmic components can leak out, eventually leading to cell death [39]. Due to their rapid action and complex resistance formation, AMPs have proven to be an excellent alternative to conventional antibiotics [40,41]. The cationic peptide lasioglossin III (LL, H-Val-Asn-Trp-Lys-Lys-Ile-Leu-Gly-Lys-Ile-Ile-Lys-Val-Val-Lys-NH_2_) is isolated from the venom of the bee *Lasioglossum laticeps* and belongs to the group of AMPs. It is an alpha-helical peptide with a hydrophobic and a hydrophilic side [39,40,42]. It shows a high antimicrobial activity against Gram-positive and Gram-negative bacteria, such as *B. subtilises*, *S. aureus*, and *E. coli* [42], and anticancer activity against PC12 or leukemia cells [42]. Due to its activity at physiological salt concentrations, LL is a promising substance for cancer therapy [42]. The positive charge and free amine groups make the peptide suitable for loading on IONs@CMD carrier particles [31,40,42].

Although many research groups have focused on the use of particles in therapeutic settings, little is known about the influence of the coating thickness on the particle properties. Thus, a need has arisen to identify critical parameters that can influence and help to standardize the design of CMD-coated IONs, as emphasized by the EMA. Yet, in particular, a lack of knowledge exists regarding the influence on agglomeration, magnetophoretic behavior, and oxidation. [1,43]. Furthermore, it is of general interest to improve the knowledge about the application of antimicrobial peptides for targeted drug delivery. In this work, five CMD-coated IONs were synthesized with a systematic increase in the CMD quantity (6.25 g L^−1^, 12.5 g L^−1^, 25.0 g L^−1^, 125 g L^−1^, and 250 g L^−1^), which were fully characterized. The presented data provide new insights into the applicability of CMD-coated IONs for magnetically controlled drug delivery in combination with the antimicrobial peptide LL.

## 2. Results and Discussion

IONs@CMD were synthesized by applying increasing amounts of CMD to the reaction mixture: 6.25 g L^−1^ (ION@CMD6.25), 12.5 g L^−1^ (ION@CMD12.5), 25.0 g L^−1^ (ION@CMD25.0), 125 g L^−1^ (ION@CMD125), and 250 g L^−1^ (ION@CMD250). The particle properties and the influence of the coating thickness on them were analyzed in detail, providing new trends and results regarding their agglomeration behavior, magnetophoresis, oxidation, and cytocompatibility. Two different TEM techniques were used to visualize the coating. The new knowledge was used to choose the ideal ION@CMD for the drug delivery of LL. The system was analyzed for its binding abilities of the AMP and its antimicrobial properties.

### 2.1. Particle Composition

The particle composition defines the critical parameters of IONs@CMD influencing the particle size, surface properties, agglomeration, and magnetic behavior. The successful binding of CMD to the ION surface was determined by FT-IR (Figure 2a). The BIONs show a characteristic νFe-O peak at 582 cm^−1^ and additional bands at 1630 cm^−1^ and 3386 cm^−1^ that can be assigned to the νO-H vibration. These result from adsorbed water on the iron oxide surface [44,45]. The IONs@CMD have further νO-H and δO-H vibrations at 3363 cm^−1^ and νC-H and δC-H vibrations at 2922 cm^−1^ and around 1410 cm^−1^, and νC=O vibration is visible at 1593 cm^−1^. The peak at 1017 cm^−1^ is attributed to the νC-O bonds [30,44]. Increasing CMD concentrations led to higher characteristic CMD bands and a thicker CMD layer [35,46]. Das et al. confirmed this trend with ATR-IR and XPS measurements, finding that increasing the addition of the polysaccharide in co-precipitation led to a higher C–Fe ratio on the particle surface [47]. Raman spectroscopy can observe a similar trend to FT-IR but also provides information about the oxidation state of the iron oxide core (Figure 2c). The typical vibration for Fe-O in iron oxide is visible at 680 cm^−1^ [48]. Two bands are characteristic in the CMD spectrum and suggest that the (C-OH) and (C-O-C) frequencies form a broad band between 1060 cm^–1^ and 1125 cm^−1^, respectively (Appendix A) [49,50]. The intensity of the iron oxide peak decreases with the increasing CMD on the surface. In the case of ION@CMD250, the signal is significantly stronger than that of ION@CMD6.25–125 and probably blankets that of the IONs. Specifically, the iron oxide peak is composed of magnetite (660 cm^−1^) and maghemite (710 cm^−1^), depending on the oxidation state [48,51,52]. Therefore, the Raman spectra were used to calculate the magnetite content (Figure 2d, Appendix A). The BIONs have a calculated magnetite content of 15.6%. The content is consistent with those in the literature [48,52]. The magnetite content increases with the thicker CMD coating from ION@CMD6.25, with 15.6%, to ION@CMD25.0, with 47.5% (Table 1). ION@CMD125 consists of 41.6% magnetite, suggesting that a plateau was reached. The increasing magnetite content is illustrated in Figure 2d by Voigt fits, showing that the CMD coating slightly protects the IONs from oxidation [53]. The crystal structure of the IONs was determined using XRD measurements (ION@CMD12.5: Figure 2b, other particles: Appendix A). All the particles show the characteristic iron oxide reflections at 13.7°, 16.1°, 19.5°, 25.4°, and 27.7°, which can be assigned to the crystal plane of iron oxide located at (220), (311), (400), (511), and (440), according to Miller’s index [23].

Therefore, the coating does not influence the crystal structure of the particles. Furthermore, CMD leads to amorphous signals between 2° and 10°. A comparable phenomenon has been observed with silica-coated IONs [54]. FT-IR and Raman show that higher CMD concentrations increase the CMD coating thickness, protecting the magnetite from oxidation to maghemite. The particles have a characteristic crystalline iron oxide core (spinel).

### 2.2. Surface Properties

In biomedical applications, the particles are subjected to the different pH values and ionic strengths in the human body. The pH values vary from pH 7.4 in the blood and cytosol to pH 6.4 in cancer cells, pH 4–5 in endosomes and lysosomes, and pH 2 in gastric acid [55,56,57]. Consequently, obtaining colloidal stable particles over a broad pH spectrum is essential. The stability and surface charge of the particles were determined by zeta potential and DLS measurements (Appendix A). The IEP of the BIONs was determined to have a pH of 7.10, which is consistent with the literature values [31]. The BIONs show significant agglomeration (d_H_ = 503.9 nm–2042 nm) around the IEP, since they have a zeta potential of ±10 mV, making the particles unstable [58]. At pH < 6.5 and >8.5, the BIONs have a hydrodynamic diameter (d_H_) of around 100 nm. Here, the zeta potentials are in a range in which the particles are moderate to very stable. The isoelectric point (IEP) of the IONs@CMDs is between 4.6 and 1.7, shifting in the acidic pH region as the coating of the particles becomes thicker (Table 1) [35]. The charge density of the CMD-coated IONs varies with the pH by changing the degree of ionization of the CMD. With a modification degree of one carboxyl group per glucose unit, the pKa value may vary from 3.3–4.5 [59]. Since it is assumed that the number of carboxyl groups increases with the coating thickness, the trend can be explained. ION@CMD6.25 and ION@CMD12.5 have IEPs of 4.6 and 4.4, respectively. These particles agglomerate around the IEP (zeta potential of ± 10 mV). The CMD side chains’ steric repulsive forces are hypothesized to be insufficient to ensure colloidal stability around the IEP. In addition, for the low CMD concentrations, the threshold value necessary for the complete coating of the particles is not exceeded and, thus, the particles are not entirely coated [47]. Therefore, ION@CMD6.25 and ION@CMD12.5 are influenced by the BION-like properties.

Nonetheless, both particle types are in the ideal size range of <100 nm at a physiological pH, with hydrodynamic diameters of 66.3 ± 7.35 nm for ION@CMD6.25 and 87.1 ± 11.0 nm for ION@CMD12.5. Thicker CMD coatings lead to a good colloidal stability over the broad pH spectrum and no agglomeration (d_H_ < 100 nm) at the IEP at pH 3.9 for ION@CMD25.0, pH 2.4 for ION@CMD125, and pH 1.73 for ION@CMD250 (Appendix A). However, agglomeration with a plate-like shape is visible in the SAXS curve of ION@CMD250 at pH 7, while the primary particle size is determined to be 20 nm (Appendix A). All the particles have a negative surface charge from a pH value of ~4.6 and higher, making them a promising material for nanomedicine. It is well known that negatively charged particles experience prolonged blood circulation, increased cellular uptake, and a lower cytotoxicity [35,60]. Furthermore, they have the potential to adsorb the positively charged LL.

### 2.3. Particle Size

The size and agglomeration behavior of the nanoparticles are crucial criteria for their applications in nanomedicine. Here, adverse effects and the blood circulation time depend strongly on the hydrodynamic diameter, shape, and surface chemistry [24,25]. An ideal particle diameter lies between 10 and 100 nm for intravenous injection, avoiding extravasation and rapid elimination by the kidneys (<10 nm) or opsonization and removal from the bloodstream by the macrophages (>100 nm). For the crossing of the blood–brain barrier, similar sizes are favorable [61]. Popovtzer et al. showed that gold nanoparticles with a size of 20 nm accumulate in the brain within two hours post-injection [62]. In addition, a large specific surface area and functional groups are desirable for high drug loadings [23,24]. Before the influences of various simulated body fluids can be analyzed, it is necessary to determine the diameter of the iron oxide core and the coating thickness. The magnetic core size is calculated from the data of the XRD spectra using the Scherrer equation (SI-Equation (S3), Table 1). The BIONs’ size of 8.8 ± 0.9 nm is comparable with the literature data [23,31]. The ION core decreases from ION@CMD6.25, with 10 ± 0.5 nm, to ION@CMD250, with 0.6 ± 0.3 nm. The larger diameter of ION@CMD6.25 compared to the BIONs can be explained by the higher reaction temperature (85 °C vs. 25 °C) [63,64]. Increasing CMD concentrations lead to smaller core sizes. The reason, therefore, lies in the conventional nucleation theory [63,65]. With higher amounts of polymer added, the CMD is more likely to meet the freshly formed crystals and stop the nuclei growth. Thus, more CMD accumulates on the particle surface [30]. Even with extended measurement times, the thicker the CMD coating is, the higher the signal-to-noise ratio will be during XRD analysis (Appendix A). For that reason, the inaccuracy of the calculated diameters slightly increases. The average d_TEM_ of 8.7 ± 1.6 nm for the BIONs fits the literature and is comparable to the d_Scherrer_ (Table 1, Appendix A) [23,31,48]. ION@CMD6.25 and ION@CMD12.5 reach a similar size of 11 ± 2.1 nm and 11 ± 2.8 nm. Thus, ION@CMD12.5 accumulates in smaller clusters (Appendix A). With the increasing input of CMD in the co-precipitation reaction, the size of the particles decreases. ION@CMD250, 125, and 25 have smaller diameters than the BIONs, with measurements of 6.3 ± 1.2 nm, 7.6 ± 1.6 nm, and 8.0 ± 1.7 nm, respectively. This trend is consistent with the decrease in the d_Scherrer_. ION@CMD25, 125, and 250 also tend to collect in larger clusters.

This phenomenon can be attributed to the fact that smaller particles have a higher surface-to-volume ratio and high surface energy. Accordingly, they agglomerate to reduce this surface energy [46]. Since the diameters of the BIONs determined by XRD and TEM coincide, the thickness of the CMD coating was calculated by determining the difference between d_TEM_ and d_Scherrer_.

It is necessary to emphasize that liquid conditions can influence the properties of the polymer coating. In agreement with the IR and Raman spectra, it can be observed that the CMD layer increases from ION@CMD6.25 to ION@CMD250. With a Δd of 5.68 nm, ION@CMD250 has the thickest coating. The thicknesses of ION@CMD125, 25, 12.5, and 6.25 are 2.82 nm, 2.42 nm, 2.35 nm, and 0.77 nm, respectively. The particles of ION@CMD12.5 appear to have a suitable layer thickness, because they fall in the size range of 10–100 nm and do not collect into larger clusters. Imaging with high-angle annular dark-field scanning transmission electron microscopy (HAADF-STEM) provides further insight. The particles show atomic lattice planes (Figure 3, Appendix A). With differential phase-contrast (DPC) imaging, the nanoparticle core can be easily distinguished from the polymer due to the fact that the sample deflects the beam significantly in both DPCx and DPCy. Again, in the experiments, atomic lattice planes were observed in the core, confirming the presence of a crystalline core and amorphous exterior polymer coating. The CMD coating showed sensitivity to the scanning electron beam in the STEM mode, leading to the growth of the polymer layer and causing unforeseen artifacts (Figure 3b orange circle). To ensure that the CMD layer’s visual representation was minimally influenced by the electron beam and, hence, artifact-free, the TEM mode was coupled with a direct electron detector to quantify and control the imminent electron dose. To conduct the imaging in this manner, the dose rate was kept under 15 e^-^/Å^2^/s by monitoring the dose rate output according to the direct electron readout and adjusting the monochromator accordingly. By applying a slight defocus, the CMD layer became more apparent compared to the iron oxide core (Figure 4).

Our detailed analysis of the low-dose TEM micrographs showed a core diameter of 6.6 nm and a medium coating thickness of 1.4 nm for ION@CMD250, while ION@CMD12 had a core size of 9.7 nm and a CMD layer of 2.1 nm, and ION@CMD6.25 showed values of 8.9 nm and 2.6 nm (Appendix A). These data fit the trends observed previously. The size of the dried, not perfectly round agglomerates lies between 92 and 101 nm for ION@CMD250, 296 and 404 nm for ION@CMD12.5, and 146 and 275 nm for ION@CMD6.25 (Appendix A). The agglomeration behavior, which is highly dependent on the medium, is essential for analyzing the applicability of drug delivery systems [40,59]. Figure 5 shows the size distributions of the BIONs and IONs@CMD in water (pH 7.4), PBS (pH 7.4), and human plasma, determined by DLS. For all the media, the BIONs have a higher tendency to agglomerate than the CMD-coated particles. In water, at a pH close to the IEP, the BIONs experience a lack of electrostatic repulsion due to their unfunctionalized surface. They form agglomerations of around 57.9 particles, with a d_H_ of 503 ± 10.5 nm. The CMD coating positively influences the particles’ stability. Therefore, all the IONs@CMD experience less agglomeration (Figure 5) [35,46]. For ION@CMD12.5, 25, 125, and 250, in ascending order, the agglomeration rises with the increasing coating thickness to 87.1 nm (7.99× d_TEM_ of one particle), 94.2 nm (11.8× particles), 162 nm (21.5× particles), and 200 nm (31.6× particles, Figure 5a, Appendix A). The only exception is the particles with the thinnest CMD coating, with a hydrodynamic diameter of 137 nm (12.6 particles), being higher than ION@CMD12.5. Das et al. also observed increasing DLS sizes when the Fe/CMD ratio was reduced to a limit where the particle surface is no longer entirely coated (17.143:1 to 12.000:1) [47]. The stabilizing effect of CMD can be explained by the negative surface charge and resulting repulsion of the particles with the same net charge. All the coated particles have a zeta potential of <−15 mV at a pH of 7–7.4, reflecting moderately stable particles (Appendix A) [58,66]. The effect of the electrolytes was studied in 50 mM PBS (pH = 7.4). The high electrolyte concentration of the medium stimulates the agglomeration of the BIONs. With a diameter of 1902 nm, which is 3.78× (d_H,PBS_/d_H,H2O_) greater than that in water, the uncoated particles reach agglomerate sizes in the µm range (Figure 5b, Appendix A) [31,67]. CMD-coated particles are stably dispersed in biological media throughout a wide range of pH and ionic strength values [35,47,68]. Therefore, CMD coating also leads to stabilization in PBS buffer. ION@CMD6.25, 12.5, 25, 125, and 250 have hydrodynamic diameters in PBS of 165.7 nm (d_H,PBS_/d_H,H2O_ = 1.21×), 157.6 nm (1.81×), 73.6 nm (0.78×), 34.5 nm (0.21×), and 69.3 nm (0.35×), respectively (Appendix A).

With higher CMD amounts, the agglomeration in PBS is less significant than that in water. Almasri et al. reported that negatively charged phosphate ions can be adsorbed more strongly on negative surfaces in the presence of other anions [69]. The ionic effects enhance the stability of the particles. PBS buffer at pH 7.4 corresponds to the cytosol conditions in the human body. The good stabilization renders the IONs@CMD favorable in drug delivery systems. For this application, the CMD-coated particles are analyzed in human serum (Figure 5c). The BIONs and IONs@CMD show a reduced agglomeration behavior due to the increased viscosity (4 mPa·s) and the potential stabilizing biomolecular corona formation [31,70,71,72]. The BIONs are agglomerated 0.87 times less than they are in water, with a diameter of 442 nm, while the increasing CMD amounts lead to up to 0.13 times less agglomeration for ION@CMD250, with 25.8 nm. The decrease in the hydrodynamic diameter in human plasma with the increasing CMD coating correlates with the charge of the particles. Tekie et al. showed that particles with increasing carboxyl groups on the surface and, correspondingly, more negative charges are very stable in serum conditions [73].

In summary, a reduction in the magnetite size with increasing CMD was detected. The particles are more stable than the BIONs in all the media used and, thus, show advantageous properties for their use as drug delivery systems. A trend can be seen with the PBS and human serum, where the particles with thicker coatings form smaller agglomerates than those with thinner layers.

### 2.4. Magnetization

SQUID and STEP technologies analyze magnetic behavior depending on the particle composition, size, surface properties, and agglomeration. The BIONs and IONs@CMD show the characteristic sigmoidal curve typical of a superparamagnetic behavior (Figure 5d) [2,23]. Superparamagnetic particles are only magnetic in the presence of an external magnetic field and have no remanence [74]. The ideal curve shape is simulated using the LangevinMod fit (SI-Equation (S4)). With less CMD coating, the particles’ slope deviates from the fit more significantly above a magnetic field strength of ±20 kOe. The saturation magnetization decreases with a thicker layer and a smaller iron oxide core. ION@CMD6.25, with a maximum saturation magnetization of ±60 emu g^−1^, is comparable with the BIONs (±67 emu g^−1^), while ION@CMD20 has a significantly lower saturation magnetization of only 25 emu g^−1^. Unterweger et al. observed a decrease in the saturation magnetization in the case of dextran-coated IONs [75]. The hydrodynamic diameters of the particles influence the magnetophoretic behavior in water to a greater extent than the magnetization (Figure 5e). Agglomeration increases the sedimentation velocity, therefore explaining the increasing sedimentation rate [76]. The BIONs sink the fastest in a magnetic field compared to the coated particles, with a sedimentation velocity of 1152 µm s^−1^ (Appendix A). Similar to the DLS measurements in water, the increasing stabilization of the particles can be seen from ION@CMD250 to ION@CMD12.5. ION@CMD250, 125, and 25 settle at 160 µm s^−1^, 90.2 µm s^−1^, and 30.7 µm s^−1^, respectively. The most stable particles, ION@CMD12.5, have a sedimentation velocity of 18.4 µm s^−1^. At the lowest coating thickness, the ION@CMD6.25 particles sink faster at a rate of 78.9 µm s^−1^, proving the agglomeration behavior. Previous studies of oleate-coated IONs have shown the effect of a distinct acceleration of the magnetic field on the sedimentation velocity [77].

In conclusion, all the particles exhibit superparamagnetic properties, with lower magnetizations detected with higher CMD coatings due to the smaller iron oxide cores and higher polymer mass. The agglomeration of the particles strongly influences the magnetophoretic sedimentation rate. All the IONs@CMD, compared with the BIONs, show a slower sedimentation speed because of their better stabilization. The ION@CMD12.5 particle type proved to be the most stable.

### 2.5. Cytocompatibility

The cytocompatibility of the BIONs and the various IONs@CMD was analyzed in direct contact with the HUASMCs after one and three days (Figure 5f). All the particles show more than 70% viability at the tested concentrations, the threshold suggested by ISO-10993 for cytocompatibility. Furthermore, there are no visible influences on the cell morphology and proliferation compared to the negative control (Appendix A). The data fit several cell viability tests of IONs in the literature. For example, Kumar et al. showed a good cell viability for MCF-7 and HepG2 cells, with values between 0.06 and 1.00 g L^−1^ for superparamagnetic IONs and folic-acid-coated ones [78]. Zhang et al. found a slight decrease of up to 20% in the cell viability of smooth muscle cells incubated with three different coated IONs (DMSO, APTS, GLU) [79]. The excellent cytocompatibility of IONs@CMD makes them suitable for application in drug delivery.

Characterizing the different CMD thicknesses provided us with a better understanding of the size intervals, stability in physiological-like media, controllability in magnetic fields, and the negative surface charge. ION@CMD12.5 stands out as the particle type with the best properties due to the ideal diameter d_TEM_ of >10 nm, the colloidal stability in water and human plasma with a hydrodynamic diameter in the range of 10–100 nm, and the good magnetization. In this context, we decided to examine the loading of LL on the ION@CMD12.5 particles.

### 2.6. Electrostatic and Covalent Binding of Lasioglossin

For the generation of a new and efficient drug delivery system, the binding of the antimicrobial peptide lasioglossin was examined by electrostatic and covalent binding. The adsorption was performed in 50 mM PBS buffer (pH = 7.4), showing a peptide loading of 0.32 ± 0.06 g g^−1^ (equilibrium concentration 3.68 g L^−1^, Figure 6a, Appendix A). Comparable to Turrina et al., where the interaction of LL with BIONs was analyzed, a substantial decrease in the loading was observed after one wash step, with a loading of 0.09 ± 0.05 g g^−1^ and 0.02 ± 0.02 g g^−1^ after the second wash step (Figure 6a) [31]. The LL, bound by weak electrostatic forces, detached from the surface as the supernatant changed with each wash step, and new equilibrium concentrations were established [31]. Although good drug loadings (24.27%) were achieved in the adsorption step, only 1.30% of LL (stock solution of 4.00 g L^−1^) remained on the surface after the two washing steps. Comparable values were also obtained by Qu et al. and Luo et al., with a maximum loading of 11.8% for 10-hydrodycamptothecin on PEG-chitosan-coated IONs and 35% for paclitaxel on magnetic colloidal nanocrystal clusters [80,81]. The peptide adsorbs quickly on the ION surface, and after only five minutes, an equilibrium is formed (Appendix A). Contrary to electrostatic bonds, covalent bonds are stronger, pH-independent, and more thermostable [82]. The following covalent binding was performed using a two-step EDC/NHS coupling protocol (Figure 6b) [83]. A peptide bond was formed between the activated carboxyl group of CMD on the particle surface and a free amine group of LL. Five concentrations of LL (0.80 g L^−1^–2.50 g L^−1^) were added, and FT-IR, DLS, and the zeta potential were used to characterize the particle–peptide complex.

The loading was determined by UV/VIS analysis of the supernatants at 280 nm and by TGA measurements. A loading of 0.55 g g^−1^ was achieved with a 0.80 g L^−1^ LL input. With a drug loading (DL) of 35.4% (280 nm), 11.2% more was loaded than the highest loading during adsorption (Table 2). From this point, increasingly higher loadings were achieved with the progressive addition of LL (Figure 6a). At the initial LL concentrations of 1.00, 1.50, 2.00, and 2.50 g L^−1^ peptide, loadings of 0.60 g g^−1^ (DL 15.0%), 0.94 g g^−1^ (DL 43.8%), 1.32 g g^−1^ (DL 52.4%), and 1.65 g g^−1^ (DL 62.3%) were achieved (Table 2). The binding efficiency varied between 28.4% and 8.07% (Appendix A). Studies on the conjugates of xylane- and dextran-coated particles with ibuprofen and naproxen achieved comparably high drug loadings (30–70 wt%) by covalent coupling [84,85]. Analogous with adsorption, the FT-IR spectra of the loaded particles show the characteristic LL bands at 1653 cm^−1^ (νC=O) and 1535 cm^−1^ (δC-N, Figure 6d). The loading was additionally confirmed and determined using TGA measurements of the dried particles (Figure 5c).

The TGA measurements of the pure peptide show a multistep breakdown process, which is consistent with the thermal behavior of freeze-dried proteins (Appendix A). At a temperature of 450 °C, approximately 20.0% of the LL is not completely burned, which is included in the calculations [86,87,88]. A constant profile with no further decrease in the weight can be seen from 300 °C until the final process temperature of 700 °C is reached. The magnetite is not completely burned, since its melting point is 1538 °C [89]. Drug loadings of 15.4%, 15.3%, 26.9%, 33.2%, and 49.7% are achieved with inputs of 0.80, 1.00, 1.50, 2.00, and 2.50 g L^−1^ LL, respectively (Table 2). The two different analytical methods vary from each other. The TGA profiles of the unbound and bound LL deviate from each other.

Higher peptide binding increases the hydrodynamic diameters, leading to a plateau of around 5 µm (1.32 g g^−1^ and 1.65 g g^−1^). The smallest hydrodynamic diameter is achieved at a loading of 0.55 g g^−1^ (96.4 nm). Here, the peptide binds with an efficiency of 82.2% (Appendix A). Compared to the adsorption, the zeta potential of the particles rises with the increasing loading (Table 2). In the experiments, charges of −29 ± 1.0 mV, 2.9 ± 1.0 mV, 2.4 ± 0.3 mV, 13 ± 0.4 mV, and 32 ± 0.9 mV were measured.

In summary, significantly higher drug loadings and a higher efficiency were achieved with the two-step EDC/NHS protocol than with the adsorptive method. The weak interactions during adsorption (electrostatic binding, van der Waals forces) are not sufficient for the efficient loading of the particles [90,91]. High-bound peptide amounts increased the hydrodynamic diameters.

### 2.7. Antimicrobial Activity

With their antimicrobial properties, ION@CMD12.5, ION@CMD@LL (ads), and ION@CMD@LL (cov) were co-incubated with green fluorescent protein (GFP)-expressing *E. coli* (BL21, amp resistance). Two different methods, namely OD600 measurements (Appendix A) and microscopy, were used to determine the antimicrobial activity (Appendix A). First, ION@CMD12.5 without the bound peptide was investigated (Figure 7a). Concentrations between 0.01 g L^−1^ and 0.40 g L^−1^ led to comparable *E. coli* growth, as was the case without the particles. The LL’s minimum inhibition concentration (MIC) lay at 1.13 to 3.70 µM [31]. Adsorbed to BIONs, the antimicrobial activity of LL could be slightly improved to 0.53 µM in previous works [31]. ION@CMD12.5@LL (ads) showed less bacterial growth with comparable LL (≤1.13 µM) concentrations. Starting at a concentration of 1.13 µM of adsorbed LL, the inhibition of E.coli could be seen for 13 h in the OD600 measurements (Appendix A). At 1.70 µM of adsorbed LL, the growth was completely inhibited. In microscopic cell counts, the complete inhibition of cell growth was evident at a concentration of 1.13 µM (Figure 7b, Appendix A). The slight difference between the two methods can be explained by the fact that the samples used for the microscopy were shaken more intensively with an overhead shaker than the linear gentle shaking of the 96-well plate [31]. Covalently bound LL affected the cell growth, starting with a concentration of 2.83 µM (OD600) or 0.40 µM for the microscopy experiment (Figure 7b).

The complete inhibition of the cell growth was seen at the 4.42 µM (OD600) and 1.70 µM bound LL concentrations, respectively. Although the MIC was slightly higher than the LL bound by physisorption, the biological activity of the covalently bound LL could be confirmed. The covalent binding of other drugs on CMD-coated IONs showed no or a slightly negative effect on the drugs’ activity [30,92].

In summary, the ION@CMD showed no or little effect on the cell growth. ION@CMD@LL (ads) and ION@CMD@LL (cov) could completely inhibit the bacterial growth at concentrations of bound LL as low as 1.13 µM and 1.70 µM.

## 3. Materials and Methods

### 3.1. Synthesis of ION@CMD

According to the Massart process, the CMD-coated IONs were synthesized by co-precipitation [30,31]. A total of 20 mL of CMD solution (250 g L^−1^ (CMD250); 125 g L^−1^ (CMD125); 25 g L^−1^ (CMD25.0); 12.5 g L^−1^ (CMD12.5); 6.25 g L^−1^ (CMD6.25); CMD sodium salt, BioXtra, Sigma Aldrich, Darmstadt, Germany, 39422-83-8); and 2.5 mL of aqueous 25% ammonium hydroxide solution (Aldrich Chemistry) were added to a 100 mL round-bottomed flask in a nitrogen atmosphere. The reaction was induced by adding 20 mL of an iron (II/III) solution (FeCl_2_·4H_2_O (1 eq., 347 mg 1.75 mmol), Emsure^TM^; FeCl_3_·6H_2_O (2 eq., 945 mg, 3.50 mmol), Fluka Sigma Aldrich, Darmstadt, Germany) to the reaction mixture and by stirring it uniformly for one hour at a temperature of 85 °C. After the completion of the reaction, the synthesized particles were centrifuged (CMD250, CMD125) at 4000× *g* for 10 min or magnetically separated (CMD25.0, CMD12.5, CMD6.25) and washed with ethanol absolute (2×) and degassed using double-distilled water (ddH_2_O, 2–3x) until a conductivity lower than 200 µS cm^−1^ was obtained. The particles were stored in an N_2_ atmosphere at 4 °C in degassed ddH_2_O.

### 3.2. Characterization

The presence of the functional groups of CMD and the presence of LL (lasioglossin III) on the particle surface were confirmed by FT-IR spectroscopy (Fourier-transform infrared spectroscopy) (Alpha II; Bruker Corporation; Billerica, MA, USA) and platinum attenuated total reflection module. A total of 3 µL (>1.00 g L^−1^) of the particle solution was measured over a wavenumber range of 4000 cm^−1^ to 400 cm^−1^ (24 scans). The background was subtracted with the software OPUS8.1 using the concave rubber band method. Each spectrum was normalized to the magnetite band at approx. 580 cm^−1^. Transmission electron microscopy (TEM) was used to determine the particles’ morphology and size. After ultrasonication, the samples (10 µL) with a concentration of 0.03 g L^−1^ were deposited onto a carbon-coated copper grid that was prepared via glow discharge and dried by blow-drying. Images at a magnification of ×120 k were recorded with the TEM JEM JEOL 1400 plus and analyzed using ImageJ software, v1.52a. At least 100 particles were measured per synthesis from a minimum of three different areas.

High-angle annular dark-field scanning TEM (HAADF-STEM) and integrated differential contrast (iDPC) STEM imaging were conducted using a Thermo Fisher Scientific Titan Themis Cubed microscope operating at an acceleration voltage of 300 kV and tuned with a monochromator and probe corrector. HAADF-STEM and iDPC imaging were conducted using Velox software (Thermo Fisher Scientific, Ma, USA). TEM imaging was conducted using the same instrument and tuned with the image corrector. The micrographs were acquired with a direct electron detector (K2, Gatan Inc., Ca, USA) using the dose fractionation method (40 micrographs taken with bursts of 0.1 s of exposure), and the electron dose was kept under 15 e^−^/Å^2^/s to ensure minimal damage due to the electron beam. The dose fractionation data were processed in Gatan Microscopy Suite 3 by importing each stack. The image stack was then subjected to 2× automated alignment procedures and then summed. To prepare the samples, a Lacey carbon 200 mesh copper grid (Agar Scientific, Essex, UK) was plasma treated using a Gatan Solarus 950 Advanced plasma system (Gatan Inc.) with O_2_ for 30 s at 65 W. The TEM grid was suspended using reverse-action tweezers and using a micropipette, and a 7 µL aliquot of 100× diluted sample was deposited on the grid and left under cover overnight to enable the droplet to evaporate. To ensure complete evaporation and to minimize imaging artefacts, the grid was placed in an opened Eppendorf tube and kept under high vacuum overnight. The hydrodynamic diameters were determined by dynamic light scattering (DLS) and the zeta potential with the Zetasizer Ultra (Malvern Panalytical) of a 1 g L^−1^ solution. For the measurements in different media (ddH_2_O pH = 1–8, 50 mM PBS pH = 7.4, human plasma (Blutspendedienst des BRK, Munich, Germany)), 1 mL of the sample was sonicated for 30 min and then placed in a cuvette (Cuvetta STD UV 4 clear side, KARTELL S.p.a.) and measured at 25 °C or 37 °C. Each determined number distribution resulted from a triple measurement evaluated with the Zetasizer software. The isoelectric point (IEP) was determined by the zeta potential measurements at different pH values (pH = 1–8). For the measurements, 800 µL of the sample was added to the flow cell (DTS1070, Malvern Instruments, 5× measured). A Boltzmann fit was used to determine the pH value at which the surface charge reached zero. By powder X-ray diffraction (XRD), the lyophilized particles (Alpha 1-2 Ldplus, Christ, −60 °C overnight in vacuum) were analyzed with the diffractometer STOE Stadi-P (flatbed measurement, molybdenum source (0.7093 Å)). The saturation magnetization of the particles was determined with the use of the superconducting quantum interference device (SQUID) magnetometer. The samples (10 mg) were fixed in the center of a small plastic tube with the adhesive Fixogum (Marabu GmbH & Co KG, Tamm, Germany) and measured with the SQUID magnetometer MPMS XL-7 (Quantum Design, San Diego, CA, USA) at 300 K with a magnetic field variation of −50 kOe to +50 kOe. Raman spectroscopy measurements were carried out using a Raman Senterra spectrometer from Bruker Optics, Germany (488 nm laser, 1 mW, exposure 10 s, 2 co-additions). The A_1g_ band of iron oxide was fitted between 600 and 750 cm^−1^ using PsdVoigt functions (Origin) to investigate the influences of the different coatings on the magnetite to maghemite ratio [48]. The thermogravimetric analysis (TGA) of the lyophilized samples was carried out using STA 449C Jupiter in a 50.0 µL aluminum oxide crucible (5 mm × 4 mm). The weight change was detected from 25 °C to 700 °C, holding an isotherm at 700 °C for ten minutes. The sedimentation rate as a function of the magnetic field was assessed with the LUMiReader (4532-123; LUM GmbH, Berlin, Germany). After ultrasonication, samples of 1.00 g L^−1^ (pH 7–7.4) were placed in contact with five stacked cylindrical neodymium boron ferrite (NdFeB) magnets (d = 12 mm; h = 2 mm, N45, Webcraft GmbH, Gottmadingen, Germany) and measured at wavelengths of 870 nm, 630 nm, and 420 nm (profile: 1000; interval: 1 s; angle: 0°; light factor: 1.00; temperature: 25 °C; magnetization 29.1–54.4 Am^2^ kg^−1^). The processing of the obtained data was performed using the software PSA-Wizard (SEPview^TM^; analysis positions: 13.0 mm, 15.0 mm, 17.0 mm, 19.0 mm).

### 3.3. Cytocompatibility

The cytocompatibility was assessed following the ISO-10993 guidelines. For the cytocompatibility assay, human umbilical artery smooth muscle cells (HUAECs, Promocell) were expanded in a culture medium consisting of DMEM supplemented with 10% fetal calf serum (FCS, Gibco) and a 1% antibiotic/antimycotic mix (ABM, Gibco). HUASMCs between passages 4 and 5 were used for the experiments. Before the experiments, the BIONs and IONs@CMD were sterilized using H_2_O_2_ low-temperature plasma and resuspended in the culture medium. After intense overhead shaking (30 min), the samples were diluted to 0.075 g L^−1^. For the experiments, the cells were seeded on 96-well plates (Greiner Bio-One, Frickenhausen, Germany) at a concentration of 10.000 cells cm^−2^ and incubated at 37 °C with 5% CO_2_ for 24 h to allow for cell adhesion. Subsequently, the BIONs and IONs@CMD suspensions were added to the cells, and the cytotoxic effects were determined after 24 and 72 h qualitatively through cell imaging using a phase-contrast microscope (BZ-X800E, Keyence, Neu-Isenburg, Germany) and quantitively using a commercial cell proliferation test (XTT; Roche, Mannheim, Germany). A culture medium served as a negative control, while a culture medium supplemented with 2% Triton x-100 (Sigma) was used as a positive control. The XTT test was performed according to the manufacturer’s instructions. Briefly, after preparing the work solution by combining the electron coupling reagent (ECR) with the XTT solution (1:50), 50 μL was transferred to each well, and the cells were incubated for two hours. The optical density of the formazan was measured at a wavelength of 450 nm and a reference wavelength of 630 nm using a spectrophotometer (Spark, Tecan, Männendorf, Swiss). An additional measure at 0 h served as a reference to exclude the contribution of the particles to the optical density. The results are presented as normalized to the optical density of the negative control.

### 3.4. Peptide Loading

*Electrostatic Binding*: For the adsorption, different LL III (Genscript) solutions of 8 g L^−1^, 4 g L^−1^, 2 g L^−1^, 1 g L^−1^, 0.5 g L^−1^, 0.2 g L^−1^, and 0 g L^−1^ (also used as the calibration line) in 50 mM of PBS buffer (pH 7.4) were mixed (1:1, total volume 400 µL) with a 2 g L^−1^ ION@CMD12.5 particle solution. The triplicates were incubated for one hour at 25 °C and 1000 rpm in a shaking incubator (Thermomixer C, Eppendorf). After incubation, the supernatant was removed for analysis via magnetic decantation for 10 min. The following two washing steps were performed with 200 µL of fresh PBS buffer (incubation: 10 min, 25 °C, 1000 rpm). After the second washing step, 400 µL of PBS was added again to reach a particle concentration of 1 g L^−1^. Before the LL content in the supernatant was determined, the samples were centrifuged for 5 min at 17,000× *g* (Centrifuge 5418, Eppendorf, Hamburg, Germany) to separate any nanoparticles. After each step, 2.5 µL of supernatant was photometrically analyzed at 280 nm using the NanoPhotometer (Implen Nanophotometer N129).

*Covalent Binding*: For the generation of a covalent peptide bond between LL III and the free carboxyl group on the surface of the ION@CMD12.5 particles, a protocol developed by Merck Millipore was used [83]. Deviating from the protocol, a particulate stock solution of 2 g L^−1^ was washed and activated and then mixed in a 600:400 ratio with the LL stock solutions (6.25 g L^−1^, 5 g L^−1^, 3.75 g L^−1^, 2.5 g L^−1^, and 2 g L^−1^). The incubation times, washing steps, and buffer compositions were performed analogously to the protocol. After the completion of the reaction, the samples were washed (2×) with Millipore water.

### 3.5. Antimicrobial Behavior

The antimicrobial behavior of the covalent and adsorptive bound LL, as well as the ION@CMD12.5 particles, was tested with (RH)_4_-GFP-expressing *E. coli* (BL21, DE3), performed analogously to Turrina et al. [31]. The ION@CMD@LL were diluted to 7.8 mg L^−1^, 5 mg L^−1^, 3 mg L^−1^, 2 mg L^−1^, 1 mg L^−1^, 0.5 mg L^−1^, 0.2 mg L^−1^, and 0 mg L^−1^. For the unloaded particles, a dilution series of 0 g L^−1^, 0.1 g L^−1^, 0.3 g L^−1^, 0.5 g L^−1^, 0.7 g L^−1^, 1 g L^−1^, 2 g L^−1^, and 3 g L^−1^ was prepared. In the experiment, both particle types were finally diluted at 1:10.

## 4. Conclusions

Five CMD-coated IONs were successfully synthesized. Higher CMD concentrations in the synthesis led to an increasing polymer layer thickness and a reduced core size. With FT-IR, Raman, and XRD, a core–shell character was determined that can reduce the oxidation from magnetite to maghemite. While all the particles exhibited superparamagnetic properties, the saturation magnetization decreased due to the reduced core size. This study provides new insights into the physicochemical characteristics, e.g., particle agglomeration, magnetophoresis, and the zeta potentials. The coating could be visualized using different TEM techniques. Between a threshold range of 12.5–25.0 g L^−1^, the CMD particles were deemed colloidally stable over a broad pH spectrum (pH 1.5–8). All the IONs@CMD exhibited a negative surface charge at physiological pH values, promoting their stability in PBS and human plasma. Moreover, the negative zeta potentials were beneficial in binding an antimicrobial peptide to the surface. All the IONs@CMD were significantly more stable than uncoated IONs and showed hydrodynamic diameters of <100 nm, fulfilling one of the most critical characteristics of drug delivery systems [23,25]. The STEP analysis demonstrated the significant impact of agglomeration on magnetophoresis (18.4–160 µm s^−1^) and the overall stability attributed to the electrostatic and static repulsion of the CMD side chains. All the particles showed cytocompatibility (>70%) over three days in smooth muscle cells. The ION@CMD12.5 particles showed ideal properties due to their ideal diameter d_TEM_ > 10 nm, colloidal stability in water and human plasma, and good magnetization, making these nanocomposite materials an excellent candidate for magnetically controlled drug delivery.

The ION@CMD12.5 materials were suitable for both the adsorptive and covalent binding of therapeutic peptides. Significantly higher drug loadings (up to 62%) and an excellent efficiency (82.3%) were obtained with EDC/NHS coupling compared to the adsorptive method (DL 24%). The weaker electrostatic interactions were insufficient for efficient particle loading (28% efficiency). The DLS, zeta potential, FT-IR, and TGA measurements also proved the peptide loading capacity. Although the particles tended to agglomerate with increasing LL loading, an optimal hydrodynamic diameter was achieved at a drug loading of 0.55 g g^−1^. The antimicrobial experiments showed that the unloaded particles had little or no effect on the cell growth. ION@CMD@LL (ads) and ION@CMD@LL (cov) have MICs of 1.13 µM and 1.70 µM, respectively. These experiments demonstrated that the IONs@CMD can be successfully combined with antimicrobial peptides, producing an inexpensive and effective drug carrier.

## Figures and Tables

**Figure 1 ijms-23-14743-f001:**
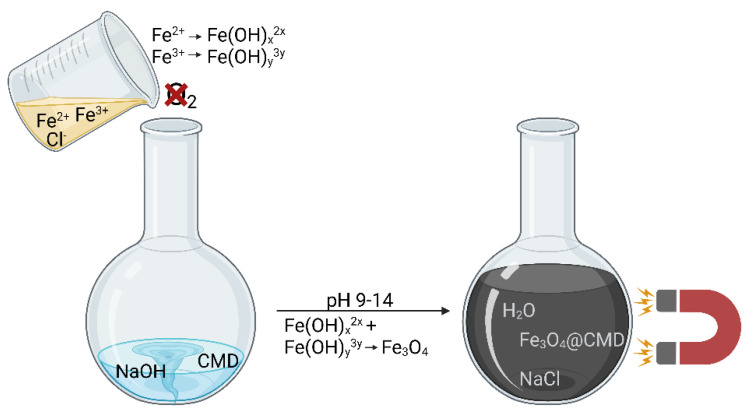
Schematic illustration of ION@CMD synthesis by the in situ co-precipitation technique according to Massart. It was created with BioRender.com.

**Figure 2 ijms-23-14743-f002:**
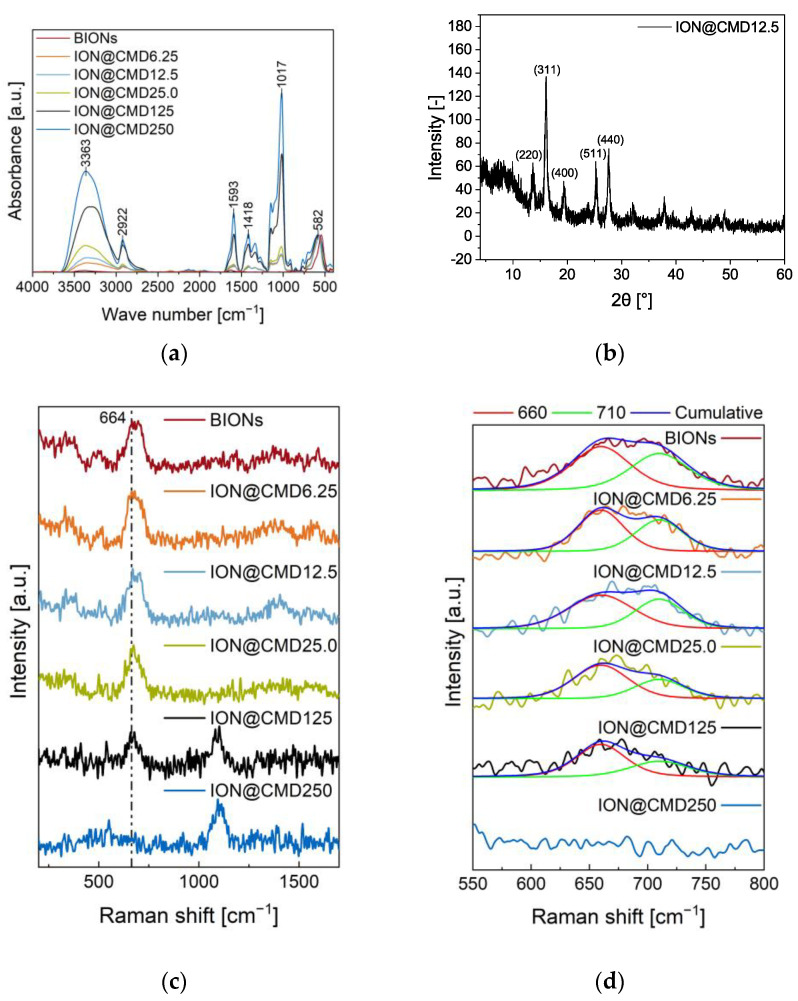
FT-IR Spectra of synthesized IONs@CMD and BIONs at wavenumbers between 4000 and 450 cm^−1^ (**a**). All spectra were normalized on the magnetite band around 582 cm^−1^. Raman spectra of ION@CMDs and BIONs (**b**). The fit of the A_1g_ band is between 600 and 750 cm^−1^ using Voigt functions in Origin (**c**). X-ray diffractogram of ION@CMD12.5 (**d**).

**Figure 3 ijms-23-14743-f003:**
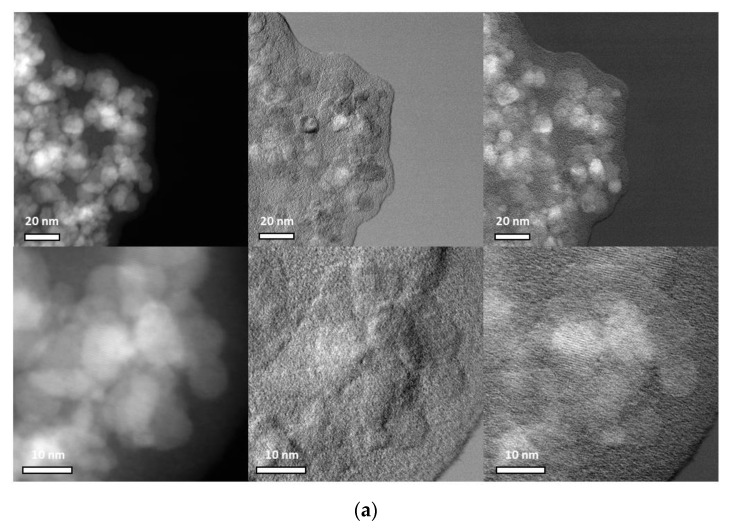
HAADF-STEM micrographs indicate the nanoparticle core with an amorphous coating for (**a**) ION@CMD6.25 and (**b**) ION@CMD250 on the left. The picture in the middle shows DPCx (A–C), and the right one shows DPCy (B–D). The orange markers show the artifacts created by the scanning electron beam. Scale bars are inset.

**Figure 4 ijms-23-14743-f004:**
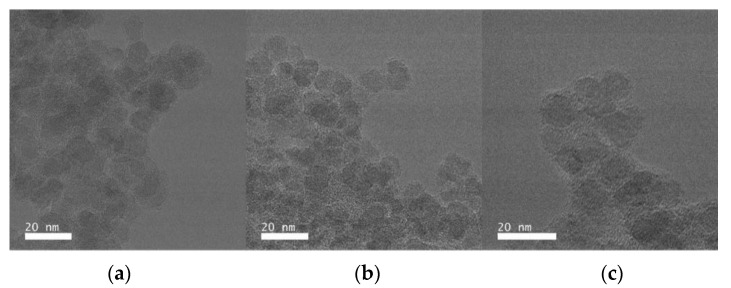
Low-dose TEM micrographs of (**a**) ION@CMD250, (**b**) ION@CMD12.5, and (**c**) IONs@CMD6.25. These micrographs were acquired using the dose fractionation method, where 40 micrographs were taken with bursts of 0.1 s of exposure and subsequently aligned and summed. Scale bars: 20 nm.

**Figure 5 ijms-23-14743-f005:**
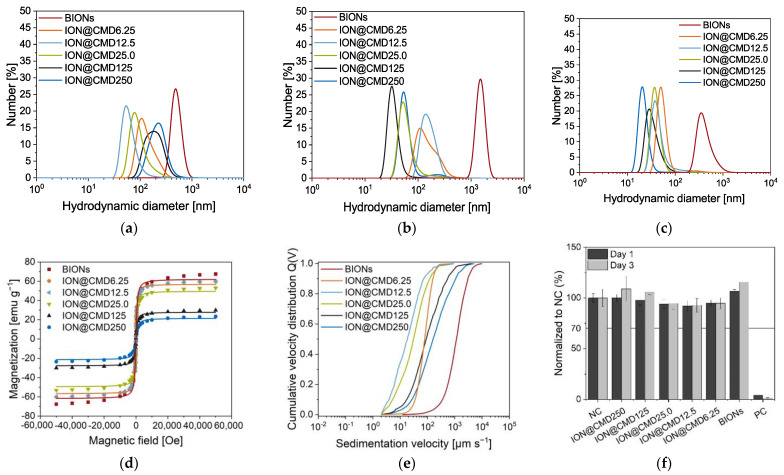
DLS measurements of BIONs and IONs@CMD in (**a**) dH_2_O pH = 7, (**b**) 50 mM PBS (pH = 7.4), and (**c**) human plasma. The equilibration time was set as 120 s. The temperature set for the water and PBS was 25 °C and the human plasma temperature was 37 °C. SQUID analysis at a temperature of 300 K using the LangevinMod fit (**d**), cumulative velocity distribution at room temperature, and (**e**) a pH ~ 7 in water. Cytocompatibility was analyzed by XTT assay with smooth muscle cells (**f**) with a negative control (NC) and a positive control (PC). Results are normalized to NC.

**Figure 6 ijms-23-14743-f006:**
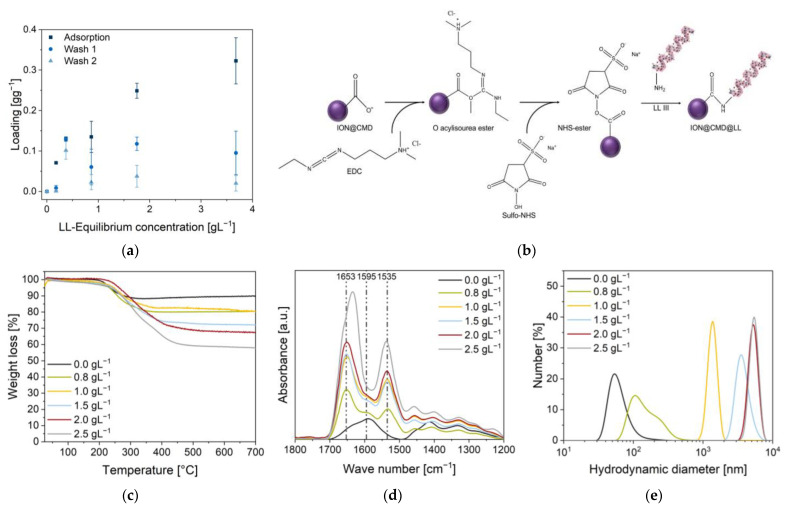
(**a**) Adsorptions of LL at pH 7.4 in 50 mM of PBS buffer and 1 g L^−1^ of ION@CMD12.5. Covalent binding of cationic peptide LL (0.0 g L^−1^–2.5 g L^−1^ input) to the surface of the ION@CMD12.5 particles. (**b**) EDC reacted with the free carboxyl group of CMD to form an unstable O-acylisourea ester intermediate. Sulfo-NHS was added to the reaction to form a more stable NHS ester, which reacts slowly with primary amines of LL to form stable amid bonds. LL was created with BioRender.com. (**c**) TGA measurements until 700 °C. (**d**) FT-IR spectra of the ION@CMD and ION@CMD@LL particles (24 scans) with labeled characteristic LL bands at 1653 cm^−1^ and 1535 cm^−1^ and the characteristic CMD band at 1595 cm^−1^. (**e**) Hydrodynamic diameters of the unloaded (ION@CMD) and loaded particles in an aqueous medium (pH = 7–8).

**Figure 7 ijms-23-14743-f007:**
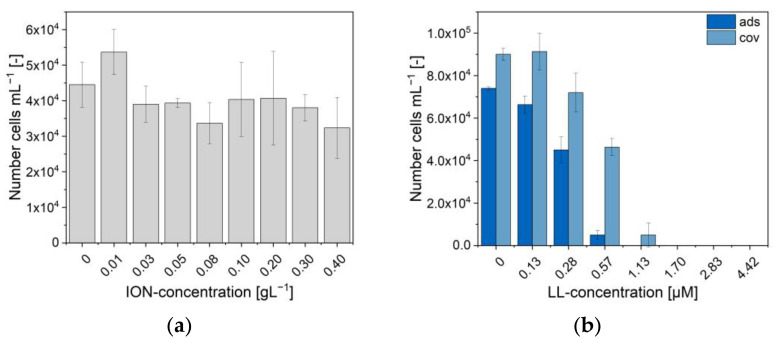
Growth of *E. coli* (BL21 (RH)4GFP-expressing) in M9 medium at different (**a**) ION@CMD12.5 concentrations and (**b**) amounts of the ION@CMD@LL complex obtained through adsorption and covalent binding. The concentration of ION@CMD@LL in diagram B describes the LL concentration.

**Table 1 ijms-23-14743-t001:** Magnetite contents of BIONs and ION@CMD6.25 to 125, determined by comparing the magnetite peak area (660 cm^−1^) and maghemite peak area (710 cm^−1^). For ION@CMD250, no iron oxide peak could be measured due to the thick CMD coating. The mean diameter was determined via TEM and Scherrer equation and IEPs.

Particles	Fe_3_O_4_ (%)	d_TEM_ (nm)	d_Scherrer_ (nm)	IEP
BIONs	15.6	8.7 ± 1.6	8.8 ± 0.9	7.1
ION@CMD6.25	21.0	11 ± 2.1	10 ± 0.5	4.6
ION@CMD12.5	33.9	11 ± 2.8	8.6 ± 0.1	4.4
ION@CMD25.0	47.5	8.0 ± 1.7	5.6 ± 0.6	3.9
ION@CMD125	41.6	7.6 ± 1.6	4.7 ± 0.4	2.4
ION@CMD250		6.3 ± 1.2	0.6 ± 0.3	1.7

**Table 2 ijms-23-14743-t002:** Loadings achieved by covalently bound LL. Drug loadings were calculated from the photometric and TGA data. Zeta potential sand DLS measurements were performed in Millipore^®^ H_2_O at a pH between 7 and 7.5.

LL Used (g L^−1^)	Loading 280 nm (g g^−1^)	Drug Loading 280 nm (%)	Drug Loading TGA (%)	Zeta Potential (mV)	d_H_ (nm)
0.8	0.55	35.4	15.4	−29 ± 1.0	96.4 ± 45.7
1.0	0.60	35.0	15.3	2.9 ± 1.0	1599 ± 401
1.5	0.94	43.8	26.9	2.4 ± 0.3	4662 ± 797
2.0	1.32	52.4	33.2	13 ± 0.4	5274 ± 11.2
2.5	1.65	62.3	49.7	32 ± 0.9	5340 ± 263

## Data Availability

The datasets generated and/or analyzed during the current study are available from the corresponding author on reasonable request.

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
