# Peer review of "Carboxymethyl-Dextran-Coated Superparamagnetic Iron Oxide Nanoparticles for Drug Delivery: Influence of the Coating Thickness on the Particle Properties"

_ijms, 2022, doi:10.3390/ijms232314743_

Round 1

Reviewer 1 Report

The manuscript entitled "Carboxymethyl-dextran coated superparamagnetic iron oxide nanoparticles for drug delivery: Influence of coating thickness on the particle properties" publishes the results of a study of magnetic nanoparticles coated with carboxymethyldextran. The topic of the article is consistent with the existing modern trends and the research is of scientific interest. The text is written logically, the narrative is consistent, the article is easy to read. The methods are described in an accessible way, the conclusions are confirmed by the results. It is recommended to accept the article after minor changes.

1) Line 64. The equation is shifted to the left, it is desirable to align it.

2) Pages 4 and 11. There is too much free space at the bottom of the page that is not occupied by the text.

3) The list of references is written in a different font, different from the font of the article.

4) Figure 2 and Table 2. Are divided into two pages.

5) Figures 2, 3, 4 and 7 are shifted to the left. It is advisable to shift them to the center of the page.

6) Lines 169 and 440. After the table, it is advisable to skip the line, separating it from the text.

7) Line 436. It looks like you didn't delete part of the text during the final editing process.

Author Response

The manuscript entitled "Carboxymethyl-dextran coated superparamagnetic iron oxide nanoparticles for drug delivery: Influence of coating thickness on the particle properties" publishes the results of a study of magnetic nanoparticles coated with carboxymethyldextran. The topic of the article is consistent with the existing modern trends and the research is of scientific interest. The text is written logically, the narrative is consistent, the article is easy to read. The methods are described in an accessible way, the conclusions are confirmed by the results. It is recommended to accept the article after minor changes.

We are thankful to the referee for the detailed review of our manuscript and the valuable requests and ideas. We appreciate the time and effort of the referee and improved our manuscript with the help of the comments.

1) Line 64. The equation is shifted to the left, it is desirable to align it.

We have addressed this comment by aligning the equation.

2) Pages 4 and 11. There is too much free space at the bottom of the page that is not occupied by the text.

Thank you for this hint. We have adjusted the graphs and tables to use the space efficiently.

3) The list of references is written in a different font, different from the font of the article.

We are thankful for this comment. We have changed the font.

4) Figure 2 and Table 2. Are divided into two pages.

Thank you for pointing this out. By adjusting the free space on pages 4 and 11, this problem was solved.

5) Figures 2, 3, 4 and 7 are shifted to the left. It is advisable to shift them to the center of the page.

Thank you. Unfortunately the graphs got shifted by the upload. But we have adjusted all of them.

6) Lines 169 and 440. After the table, it is advisable to skip the line, separating it from the text.

We have separated both lines from the table.

7) Line 436. It looks like you didn't delete part of the text during the final editing process.

Thank you for this comment. We have deleted this line.

Reviewer 2 Report

The paper entitled “Carboxymethyl-dextran coated superparamagnetic iron oxide nanoparticles for drug delivery: Influence of coating thickness on the particle properties” prepared by Turrina et al investigated five CMD-coated IONs with a systematic increase in CMD quantity. The presented data give new insights into the applicability of CMD-coated IONs for magnetically controlled 1drug delivery in combination with the antimicrobial peptide LL. The work is interesting and the results support the claims. By the way, I recommend a major revision and it is needed to address the following issue:

-The keywords are too much! Please reduce to 5

-STEM and TEM images don’t show NPs clearly, please replace images with higher resolution

-How samples were prepared for electron microscopy?

- There are many studies investigating the importance of the topic , Please add these references to your introduction and discussion parts of the manuscript and compare and bold your study novelty: https://doi.org/10.1016/j.jddst.2020.101916, https://doi.org/10.3390/pharmaceutics12080725, https://doi.org/10.1002/advs.202170049, https://doi.org/10.1016/j.mtchem.2022.101131

-please add a graphic abstract to present your goal

- The NPs stability and biocompatibility need to investigate by the authors

Author Response

The paper entitled “Carboxymethyl-dextran coated superparamagnetic iron oxide nanoparticles for drug delivery: Influence of coating thickness on the particle properties” prepared by Turrina et al investigated five CMD-coated IONs with a systematic increase in CMD quantity. The presented data give new insights into the applicability of CMD-coated IONs for magnetically controlled 1drug delivery in combination with the antimicrobial peptide LL. The work is interesting and the results support the claims. By the way, I recommend a major revision and it is needed to address the following issue:

Thanks to the referee for the detailed review of our manuscript and the valuable input and ideas. We appreciate the time and effort of the referee and improved our manuscript with the help of the comments.

We are thankful for the remark about the English. One of our authors was born in Ireland and is a native English speaker. We have double-checked the manuscript, and the English language should be fine now.

-The keywords are too much! Please reduce to 5

We are thankful for this advice and have reduced the keywords to 5.

-STEM and TEM images don’t show NPs clearly, please replace images with higher resolution

Thank you for these remarks. We used HAADF-STEM and low-dose TEM micrographs to visualize both the core and the coating. The coating visualization was impossible with standard TEM imaging (images can be found in the SI) and was not done before. Both technics produce micrographs with a slightly different might unfamiliar look. Yet we could even visualize the atomic lattice planes, which shows that the pictures are sharp. Of course, the coating lies on the iron oxide cores; therefore, the iron oxide core is slightly less sharp on the edges. This effect cannot be overcome with new micrographs.

-How samples were prepared for electron microscopy?

We are thankful for these questions. You can find a detailed description of the sample preparation in the Material and Methods chapter at 3.2 Characterization:

To prepare the samples, a Lacey carbon 200 mesh copper grid (Agar Scientific) was plasma treated using a Gatan Solarus 950 Advanced plasma system (Gatan Inc.) with O2 for 30 s at 65 W. The TEM grid was suspended using a reverse action tweezers and using a micropipette, a 7 µL aliquot of 100x diluted sample was deposited on the grid and left under cover, overnight for the droplet to evaporate. To ensure complete evaporation and to minimize imaging artefacts, the grid as placed in an opened Eppendorf tube and kept under high vacuum overnight

- There are many studies investigating the importance of the topic , Please add these references to your introduction and discussion parts of the manuscript and compare and bold your study novelty: https://doi.org/10.1016/j.jddst.2020.101916,  https://doi.org/10.3390/pharmaceutics12080725,  https://doi.org/10.1002/advs.202170049,  https://doi.org/10.1016/j.mtchem.2022.101131 

Thank you for showing us these excellent papers. All of them offer exciting work. We added the papers of Barani et al. and He et al. to our introduction in lines 49 to 51, where we give examples of the application of nanoparticles in nanomedicine. We really enjoyed reading the other two papers as well. Even though using non-conjugated polymers with intrinsic luminescence and ionically crosslinked complex gels loaded with oleic acid-containing vesicles is fascinating, we could not find an excellent context to discuss these works with iron oxide-based magnetic drug delivery. Since we already have many references, we were told by the editor only to use the relevant ones for the content.

-please add a graphic abstract to present your goal

 Thank you for pointing this out. We added the graphical abstract directly at the beginning.

- The NPs stability and biocompatibility need to investigate by the authors

We are thankful for this comment. We coincide that nanoparticle stability and biocompatibility are essential topics we plan to address in our future work. With this manuscript, we mainly focused on detailed particle characterization and analysis of drug binding and drug activity. All of these topics are discussed in depth, including colloidal stability. The cytocompatibility tests in our manuscript already give information about the non-toxicity of our particles to human cells following ISO-10993 guidelines. Two completely new topics could overload our manuscript, and as we have not established analytics for this in our laboratory, the experiments would significantly exceed revision times of the journal IJMS. However, this topic is definitely of great importance and will be addressed by us in the future. We thank you for your farsightedness and thank you for your review.

Round 2

Reviewer 2 Report

Accept